# Theoretical Reflections on Reductionism and Systemic Research Issues: Dark Systems and Systemic Domains

## Gianfranco Minati

Italian Systems Society, 20161 Milan, Italy; gianfranco.minati@airs.it

**Abstract:** In this article, we explore some theoretical issues related to reductionism and systems. Fundamentally, reductionism neglects that a system can acquire properties. Among various possible reductionist approaches, we consider the reduction of sufficient conditions to necessary conditions in systems, the reduction of emergence to functioning, and the general linearizability of non-linear systems. Furthermore, we consider the reductionistic deductibility of the macroscopic from the microscopic (as a matter of scalarity without intermediary emergence). We examine "reductionistic interacting" as it relates to multiple sequenced interactions being zippable into a single interaction. We consider the theoretical dynamic mixed usage of reductionism and non-reductionism. We then elaborate on theoretical systemic issues around opaque dark systems (as non-evident systems requiring both change in scale and change sequences). We investigate how a phenomenon can be improperly modeled as a system. This is often undertaken for the convenience of an observer (who takes advantage of the readily available approaches and models). We elaborate on the interdependence and possible equivalence of these phenomena's theoretical incompleteness and the logical openness of their modeling. We also consider the theoretical issue of systemic domains as space. Here, an entering entity only has access to certain actions and degrees of freedom due to the predominance of a previous systemic phenomenon. We conclude by considering the centrality of theoretical research in systems science.

**Keywords:** coherence; emergence; incompleteness; reductionism; quasiness; structural dynamics; systemic domain





## 1. Introduction

The purpose of this article is to elaborate on some theoretical objections against the misdirected use of reductionist approaches and narratives. Their consequences often have the detrimental effect of oversimplifying a problem in general, and even more so when we are confronted with highly complex and interconnected structures such as collective, ecological systems or the economy, for example. This article also elaborates on some unusual systemic theoretical issues such as dark systems, reverse reductionism, and systemic domains.

The theoretical nature of this elaboration relates specifically to (a) abstract systemic representations allowing for the identification of models, rules, and simulations, and (b) the properties suitable for application, generalization, and research approaches.

The theoretical nature is particularly suitable for dealing with the systemic complexity, identifying, for example, its emergent, non-linear, multiple nature, having high sensitivity to initial conditions that a macroscopic, simplifying, reductionist approach assumes as neglectable, approximal, or ignorable. This assumption often occurs because users, observers, and operators do not have a theoretical understanding of the complexity involved.

In Section 2, we consider the generic concept of reductionism.

In Section 3, we consider this concept's theoretical aspects. We deal specifically with (1) the reduction of sufficient conditions to necessary conditions, (2) reducing emergence to functioning, (3) the general linearization of non-linear systems, (4) hierarchical considerations related to microscopic and macroscopic levels (specifically whether the latter

are deductible from the former), neglecting any possible emergence between levels; and (5) reductions that putatively occur when different interactive processes are assumed to be zippable into a single process; (6) reductionism in social systems; and (7) reverse reductionism, that is, cases where a phenomenon is forcibly approximated or modeled as a system when it is not.

In accordance with the topics of this Special Issue, in the following three sections, we consider theoretical issues about systems science other than reductionism.

In Section 4, we consider the theoretical research issue related to opaque dark systems. Regarding opaque dark systems, we consider cases where the systemic nature of phenomenological dynamics and properties are hidden (due to, e.g., the assumed level of description and scalarity, and the models an observer takes into consideration).

In Section 5, we elaborate on theoretical incompleteness and logical openness. We also discuss their interdependence and related models.

In Section 6, we examine the concept of a *system domain*. This relates metaphorically to a *systemic field* where interacting elements are induced to behave like components of a predominant system.

In the conclusion, we mention how systems research should not exhaust itself creating models and simulations, but consider theorizations mixing theoretically symbolic, non-symbolic, and data-driven approaches, allowing for soft theorization and incomplete theorization. Researchers should not just pursue the (funded) decided objectives. They should, instead, pay due attention to pertinent theoretical issues.

## 2. Reductionism

Ontological reductionism relates to a whole consisting of a minimal number of parts. Methodological reductionism relates to the general possibility of an explanation of the whole in terms of ever-smaller entities [1].

Reductionism must be distinguished from a scientific method which considers quantitative aspects, that is, the collection of measurable quantities. For example, when dealing with an apple, we consider measurable variables such as its acidity (pH), color, density, size, hardness, shape, odor (odorimetric unit), surface area, volume, and weight as sufficient to take appropriate approaches, e.g., conserve and cook it. Considering measurable aspects is not a reductionist but a scientific approach. In the case of an apple, reductionism lies in considering it as a set sum of the elements that compose it, such as molecules, without considering, for example, the structures and their changes, coherence, internal processes, and local changes, e.g., apple ripening, molecular aggregations, context sensitivity, adaptability, maturation, or rotting over time: That is, neglecting that the apple is a system having the property to acquire properties.

According to Russell Ackoff [2], reductionism is a doctrine defending the idea that all entities and events (plus their properties) are made up of ultimate and indivisible elements. Systems are then made up of the sum of their parts.

In reductionist approaches, non-linear interactions and emergence processes are neglected. Complex properties are taken to be non-complex; they are treated as being completely computable and foreseeable. This is the matter of non-systemic uses of systems (reducing to objects possessing proprieties) [3].

Reductionists treat the *acquired* systemic properties as non-systemic properties. Acquired properties include traffic patterns and behaviors, whirlpooling, bird flocks, the weather, swarm intelligence, the behavior of a double pendulum, and various chaotic phenomena (evolving systems that are extremely sensitive to the initial conditions). In other words, reductionists treat acquired properties as *possessed* properties (e.g., age and weight).

In the philosophy of science, theoretical reductionists maintain that theories, entities, or laws from previous scientific theories can be logically derived or deduced from newer, broader scientific theories. This is purported to allow us to understand the relevant theories, entities, or laws as more basic or elementary. So, Newtonian mechanics, for example, should be derivable from special and general relativity. There is also an assumption that

scientific theories can be reduced to more basic scientific theories (e.g., biological theories being reduced to physical theories). However, this kind of reductionism is premised on neglecting the intrinsic multiplicity of complex systems (in which emergent, self-organizing, and chaotic phenomena occur, as introduced below). Multiplicity is related to both (a) the different possible non-equivalent levels of representation and (b) multiple interactions and multiple roles for component parts (as seen in ecosystems) [4–6].

We will elaborate on how multiple systems are intrinsically incomplete and therefore logically open. They require multiple non-equivalent models. Well-defined, monolithic systems include electronic devices. In contrast, multiplicity makes multiple systems adaptive, robust, and tolerant to structural perturbations and changes. An example is a structurally changing whirlpool, which can resume its dominant specifying properties despite perturbations.

## 3. Theoretical Aspects of Systemic Reductionism

We now elaborate on some aspects of systemic theoretical reductionism as it relates to systemic properties and their acquisition processes. We will deal specifically with theoretical aspects of systemic reductionism, its recurrence, and dual probable inevitability.

Note that we are not concerned with a simplistic notion of reducibility (as identifiable in Simplicio's famous defense of the Ptolemaic worldview) [7]. Moreover, a simplistic distinction between reductionism and non-reductionism itself seems reductionistic. The boundaries are not clearly identifiable. This is exemplified in concepts like the following: (1) a quasi-system (where the predominance of a systemic nature is only present in variable percentages) [8] (pp. 155–157), (2) theoretical incompleteness (understood as the inherent incompleteness necessary for emergence) when phenomena are incomplete enough to permit the establishment of a space of equivalences constantly incomplete, in that no specific single order or structure predominates [8], (pp. 116–122), [9] (see Section 5), and (3) multiple systems (e.g., when a system's variables are also other systems' variables) [10]. The introduction of these concepts calls into question our classical understanding of systems (as analytically identifiable and distinguishable from non-systems).

We mention how a quasi-system may be defined by its inhomogeneous possession of systemic properties. For instance, in ecosystems, different levels of openness are possible depending on the spatial location, e.g., when some areas are icy or shielded from light. Furthermore, a quasi-system may be open on the account of different variable aspects, such as being open to energy but not to information, or having such openness at different levels over time. A further example of a quasi-system is a biological system with multiple dynamically interacting co-occurring pathologies, varying its systemic multiplicity: see, for instance, [11].

Furthermore, a collective system may be able to assume intelligent behavior or acquire collective intelligence, and only in the face of specific events.

There may be reason to doubt the reducibility of (1) quasi-systems and multiple systems to systems, (2) the incomputable to the computable, and (3) theoretical incompleteness to completeness (related to completable incompleteness or neglectable incompleteness).

We are interested in how systemic quasiness supposedly relates to entities having (1) levels of tolerance regarding a temporary loss of systemic interactions, (2) levels of instability, (3) irregular collapse alternations, and (4) coherence recoveries. These typically apply to collective systems, which partially dissolve and recompose over time in response to external perturbations. Other examples of such systems include self-organizing non-linear systems like, in fluid thermodynamics, whirlpools and so-called Rayleigh–Bénard convection, when a fluid heated from below develops a regular pattern of convection cells.

The classical approach is based on looking for a unique and best-performing model. However, this style of reductionism is called into question by the intrinsic multiplicity and quasiness of complex systems.

Part of our focus will be the conceptual framework of Dynamic Usage of Models (DYSAM) (introduced in [12] (pp. 64–70) and elaborated in [8] (pp. 201–204)). Here, the emphasis is on multiplicities of phenomena (e.g., complex phenomena) that are inex-

haustible using a single approach or level of description. The general idea is as follows: "In these cases, the goal is not to find the 'best' and 'unique' approach but to use different approaches together, in such a way as to reproduce the coherent evolutionary multiplicity of real processes. This is a step towards understanding the multiple and dynamical unity of science glimpsed by von Bertalanffy" [8] (p. 9).

Some of DYSAM's theoretical roots can be identified in the Bayesian method. Following Bayes' theorem, one employs a statistical treatment based on the "continuous exploration" of events occurring in the environment under study. So-called ensemble learning algorithms are also important. They use traditional machine learning algorithms to generate multiple basic models. These are then combined into an ensemble model, which usually performs better than a single model. Another root is evolutionary game theory (based on von Neumann's "minimax theorem" [12] (pp. 64–70)).

Along with incompleteness, multiplicity, and quasiness, we are also concerned with contextual occurrences of linearity and non-linearity, involving levels of non-linearity related to networks and their properties; neural networks and their architectures; layers and weights; and metastructures (structures whose elements are also structures) [13]. When dealing with interactions between different structured coherent domains, such as those within liquids and magnetic materials, metastructures may come into play [14]. The interest in the theory of metastructures has arisen after the introduction of so-called mesoscopic physics [15]. Mesoscopic variables relate to an intermediate level between the micro- and macro-scale, in which the micro-scale is not completely neglected when adopting macro-scale levels. The mesoscopic level can be considered the place of continuous negotiations between the micro- and macro-scale. At this intermediate level, a large variety of mesoscopic representations is possible, such as considering undefined numbers of possible clusters and their intra-structural properties.

A situation similar to contrasting completeness and incompleteness, single models and multiple models, occurs when we contrast objectivism and constructivism [8] (pp. 188–194). When objectivism and constructivism are mixed, contextual usage represents daily practice.

*We will conclude that a dynamic usage of reductionism and non-reductionism is desirable.* The problem lies in making one approach absolute (to the detriment of the other). We should not overlook a mixed, variable, temporary, and possibly superimposed approach.

We now consider some theoretical issues related to systemic reductionism.

### 3.1. Reducing Sufficient Conditions into Necessary Conditions

Sufficient conditions are not necessarily necessary and necessary conditions are not necessarily sufficient. Sufficiency is more powerful than necessity because something sufficient might not be necessary. However, sufficiency can be established if different conditions are added to the necessary conditions. A package of necessary conditions is invariant if not for equivalences. A package of sufficient conditions can then be composed of different conditions and equivalent conditions. For example, the property of being a rectangle is necessary to being square; however, the property of being a rectangle is not for being square. The property of being a square is sufficient for being a rectangle; however, the property of being a square is not necessary for being a rectangle.

Sufficient conditions can both fulfill the necessary conditions or be non-necessary conditions. Of course, this does not apply when necessary and sufficient conditions coincide. An example of a necessary condition is that $Q$ cannot be true without $P$ being true. Conversely, in the proposition "If $P$, then $Q$", $P$ is sufficient for the truth of $Q$. However, $P$ being false does not always imply that $Q$ is false. We can also have propositions like "If $S(\neq P)$, then $Q$" and "If $P$, then $R(\neq Q)$". It is thus possible to have both more and different sufficient conditions.

Consider in $N$ the whole integer numbers $p > 2$. Being odd is necessary for $p$ to be prime (since 2 is the only even and prime number). However, being odd, it is not sufficient in being prime; $p$ must also be solely divisible by itself. We might think of approximating

the relationship between the set of prime numbers and the set of odd numbers as an instance of reductionism.

Consider necessarily interacting, networking, and boundary conditions as sufficient for establishing the collective behavior of a system. Here, the neglect (often unknown) of modalities (e.g., establishing coherence and long-range correlations) is a case of reductionism. Considering, in general, that all necessary conditions approximate or exhaust all sufficient conditions is a case of reductionism.

Considering processes of economic growth in social systems to be instances of development is another case of reductionism (viz. reducing development to growth). We then think about how the growth process can be conceptually understood as a sequence of increments. Such increments are often taken to be repeatable without limit. But this is often assumed without considering (a) their sustainability and (b) side effects and interactions with other processes (e.g., other growth processes). Development can be understood as the process of property acquisition, e.g., coherence, in suitable growth processes and their configurations. It should be noted that good development or bad development (understood as sustainable or unsustainable property configurations of growth systems) is inevitable. Growth processes inevitably interact, even without organization or decision-making, and acquire new properties. This can give rise to unwanted properties, including environmental and social negative effects.

*3.2. Reducing Emergence to Functioning*

A device is said to be functioning when it performs the action or activity for which it was designed. This can happen when components are suitably interconnected by way of fixed networks (e.g., in mechanical, electronic, hydraulic, or thermodynamic devices). Functioning takes place when devices are suitably powered, and the components are then allowed to interact, establishing a structured system. A set of structured components takes on acquired properties to perform a certain function. If the power supply is cut, then their key properties disappear. The structured system of components degenerates into its components. Functioning is also assumed to be adjustable and repeatable.

Components of complex systems interact in multiple ways and follow dynamic local rules [16]. This is particularly noticeable in (1) chaotic phenomena (e.g., the double pendulum and atmospheric conditions [17]), (2) self-organization (e.g., whirlpools, Bènard rolls [18], and the patterns formed in the Belousov–Zhabotinsky reaction [19]), and (3) emergence (as in collective systems like bird flocks, insect swarms, ecosystems, the morphological properties of cities, and networks [20–26]).

Self-organization is the process of unstructured acquisition of the "same" property over time (as in the whirlpools and Bènard rolls mentioned above). Emergence, in contrast, is the continuous and unpredictable acquisition of multiple coherences. It involves the acquisition of multiple properties by way of undesigned interactions and structural dynamics. Such dynamics involve variation in structure between variable components. This is observable in ecosystems that exhibit multiple and continuously changing, albeit coherent, behavior. We can say that coherence replaces structure. Another example is emergent computation [27]. This occurs when it is impossible to find an algorithm capable of computing the end state of an evolution without computing all the discrete intermediary states. Neural networks and cellular automata are prototypical examples (even if cellular automata are defined as having fully deterministic evolution rules); see Section 3.5.

Complex systems can be generated via chaos, self-organization, emergence, or quantum phenomena (and their eventual combinations). An example is multiple dynamical attractors and the properties of their dynamics. This is equivalent to considering superimposed abstract spaces of multiple attractors [8] (p. 265), [28].

Note that emergent processes require incompleteness and the multiple equivalences of quasi-systems [8]. Quasiness relates to equivalences, inhomogeneity, multiplicity, non-regularity, partiality, and the dynamics of loss and recovery. A feature of quasi-systems is that they are not always the same system. In fact, they are not always systems [8] (pp. 155–161).

Quasiness represents the incompleteness of the interaction mechanisms. This incompleteness is necessary (if not sufficient) for the realization of emergent processes. Completeness extinguishes emergence, reducing it to computable dynamics [8,10]. We mention some roots of the concept of quasi, such as quasicrystals, where, contrary to crystals, the atoms are arranged in a non-deterministic, not periodic or repetitive, structure [29]. In mathematics, quasiperiodicity relates to recurrences whose periodicity has irregular or unpredictable components.

Despite the above, reductionism remains if we mistake incompleteness for completeness. This is a matter of logical closedness contrasting with logical openness (see Section 5; see also [11] (p. 75).

Stipulating finite and limited degrees of freedom allows us to mistake incomputability for computability [27]. Ignoring incompleteness, multiplicity, and quasiness (or assuming their computability) can result in attempts at the reduction of complex systems to (or their, less dramatic, approximation as) non-complex systems. This can lead to approaches that are both wrong and counterproductive, approaches that attempt to deal with nonexistent systems [3]. Complex systems cannot be regulated or decided. They can only be suitably and interactively oriented or influenced. This occurs when we act, for instance, on environmental conditions, available energy, noise, and changing initial conditions [8] (pp. 208–210).

*3.3. General Linearization of Non-Linear Systems*

It is well known that a linear function satisfies the following two properties:

i.  Additivity: *f(x + y) = f(x) + f(y)*.
ii. Homogeneity: *f(αx) = αf(x)* for any parameter α.

However, the common understanding of "linear" often equates it to proportional. An increment of *x* is supposed to correspond to an increase that is suitably proportional (even negative) to *y*. This does not apply to non-linear functions (e.g., $f(x) = x^n, f(x) = sinx$). Non-linear functions model the non-linearity of non-linear systems for which the current state and output do not linearly correspond to the previous state and input. Examples include chaotic systems (which exhibit high sensitivity to the initial conditions) and collective systems (which have vast dynamics and a variety of local non-equivalent rules).

In his seminal *General System Theory*, Ludwig Von Bertalanffy writes as follows: "Application of the analytical procedure depends on two conditions. The first is that interactions between 'parts' be nonexistent or weak enough to be neglected for certain research purposes. Only under this condition, can the parts be 'worked out', actually, logically, and mathematically, and then be 'put together'. The second condition is that the relations describing the behavior of parts be linear; only then is the condition of summativity given, i.e., an equation describing the behavior of the total is of the same form as the equations describing the behavior of the parts; partial processes can be superimposed to obtain the total process, etc. These conditions are not fulfilled in the entities called systems, i.e., consisting of parts 'in interaction'" [30] (p. 19).

Linearization is used to approximate a non-linear system as a linear system exhibiting similar behavior (especially in the vicinity of a designated point). Examples of the methods used to linearize a non-linear system are the Taylor series expansion, state-space linearization, feedback linearization, and Jacobian linearization (which involves approximating a system's behavior to its Jacobian matrix). Linearization is used to make non-linear systems tractable (as in the linearization of the Lorenz system [31]). Examples of non-linear systems acquiring non-linear properties as behaviors include chaotic systems, adaptive systems, learning systems, evolving systems, and collective systems.

Despite the above, it can be dangerously reductionistic to assume the indiscriminate generalizability of linearization without considering validity contexts and suitable levels of representation. This can hide complex behavior-characterizing aspects of the system under study. Such aspects include bifurcations, fluctuations, a high dependence on initial conditions, long-range correlations, polarization, power laws, remote synchronizations, scale

invariance, self-similarity, symmetry breaking, temporal phenomena, local synchronizations, and network properties (e.g., the occurrence of small worlds when distant neighbors can be reached from every other node via a small number of intermediate links. Some examples are the electric power networks and the neural networks of brain neurons. Linearizing complexity induces the use of linear approaches [3]. However, structural approaches to complex systems require that we consider aspects whose effects are only approximated in linearization.

### 3.4. The Macroscopic from the Microscopic

We should begin this section with some introductory definitions.

Firstly, at the microscopic level, it is possible to deal with single elements (e.g., cells, molecules, and particles) that are separable (even if we do not take their distinguishability for granted).

Secondly, the macroscopic level relates to various collections of microscopic entities characterized by their aggregations and (acquired or possessed) properties.

These two levels (microscopic and macroscopic) can be thought of as two extreme hierarchical levels [8] (pp. 38–41). That said, the scalarity is variable if not for elementary particle physics. The study of elementary particles is devoted to researching the very lowest level of the micro–macro hierarchy. In elementary particle physics and condensed matter physics, fundamental particles (i.e., subatomic particles) are not thought to be composed of other particles. This distinguishes the lowest level from supposedly higher levels.

With a reductionistic approach, the hierarchical levels between micro and macro are analytically connected in fixed (often bidirectional) ways. The reductionistic approach is largely based on the idea that such an analytical reconstruction renders the macroscopic explainable by the microscopic. The microscopic is intended to be the so-called seat of truth; it is the source of necessary and sufficient ultimate causes. However, this might only be conclusive for a specific reductionistic explicatory model employed by an observer. Reductionism assumes analytical composability and decomposability. It assumes that there are hidden causes and generative processes responsible for the formation of the macro. Correlation is often reductionistically identified alongside causation. The problem is that correlation does not imply causation (see, e.g., [32,33]). In genetic research, for example, one might consider how a high percentage of human genes can be related to cancer. This relation makes us realize that a high percentage of human genes being cancer-related is actually irrelevant, as is the related biological research [34].

In contrast, a non-reductionistic understanding takes hierarchical levels between the micro and the macro to (a) be non-reversibly emergent and (b) contain layered phenomena of emergence (as in biological systems and ecosystems) [8] (pp. 269–273). When it comes to cancer research, what we observe at a specific level (e.g., a population of cells) can be different from what we observe at another level (e.g., a single cell).

That said, the reductionist approach is suitable for entities and systems with zero or low structural dynamics. In such cases, there is often a simple and stable interscalability between the micro and the macro. An example is approximating the Earth's surface as flat when journeying over short distances.

The concept of emergence is suitable in cases where complex systems have multiple levels of emergence, that is, when emergent entities become entities of an emergent phenomenon [8] (pp. 255–260).

### 3.5. Reductionistic Interactions

Two elements are said to interact when the behavior of one element influences the behavior of another. An example is a system model consisting of ordinary differential equations where $f_1$ represents the exchange of kinetic energy between $x_1$ and $x_2$:

$$\begin{cases} dx_1/dt = f_1(x_2) \\ dx_2/dt = f_1(x_1) \end{cases} \tag{1}$$

The situation becomes more complex when, for example, there is simultaneous validity of (1) and (2):

$$\begin{cases} dx_1/dt = f_2(x_3) \\ dx_3/dt = f_2(x_1) \end{cases} \tag{2}$$

If interactions $f_1$ and $f_2$ occur simultaneously, then we have:

$$\begin{cases} dx_1/dt = f_1(x_2) \\ dx_2/dt = f_1(x_1) \\ dx_1/dt = f_2(x_3) \\ dx_3/dt = f_2(x_1) \end{cases} \tag{3}$$

One might think of a "summarized" system of interactions:

$$\begin{cases} dx_1/dt = G(x_2, x_3) \\ dx_2/dt = f_1(x_1) \\ dx_3/dt = f_2(x_1) \end{cases} \tag{4}$$

where $G$ is a composition $G = f_1 \otimes f_2$. This eventual composition is based on a reductionistic assumption related to the simultaneity of the occurrence:

$$\begin{cases} dx_1/dt = f_1(x_2) \\ dx_1/dt = f_2(x_3) \end{cases} \tag{5}$$

intended to coincide with

$$dx_1/dt = G(x_2, x_3) \tag{6}$$

where $G = f_1 \otimes f_2$. However, the time granularity is lost. In system (4), the configuration of values $dx_n/dt$ at time $t_n$ is input to the system at time $t_{n+1}$. The reductionist view assumes the validity of $G$. This dynamic is assumed to be then zippable (or, more elementarily, summable) into G.

There are several examples where such zippability is theoretically impossible [26]. As mentioned in Section 3.2, it is impossible to zip the evolution of a cellular automaton or neural network into a single algorithmic step without computing all the discrete intermediary states. One cannot compute the end state in "one fell swoop".

In a cellular automaton, which involves the simulation of a universal Turing machine that can self-replicate, the state of a cell can only depend (via a local transition rule) on the state of the cell itself, on the state of two adjacent cells, and on others depending on the dimensionality of the cellular automaton.

Considering two-dimensional cellular automata with square cell lattices, we now mention two types of neighborhood relationships, i.e., the so-called:

- Moore neighborhood, which includes, besides the cell under consideration, the eight neighboring cells that share at least one vertex with it;
- Von Neumann neighborhood, where the cell under consideration shares at least one edge with its four neighbors.

The number $N$ of different possible local transition rules increases with the number $k$ of different allowed states per cell and with the number $r$ of cells included within the neighborhood of the cell under consideration, so that:

$$N = k^q, \text{ where } q = k^r. \tag{7}$$

Here, even with small values of $k$ and $r$, $N$ can be very large. For instance, when considering a two-dimensional cellular automaton with a Moore neighborhood, where $r = 9$ and $k = 10$, we have $N = 10^{10^9}$ [35].

Reductionistic zippability also assumes that we can ignore the subsequence of effects (e.g., interacting with the single summative energy involved, which is considered

undistributed in subsequent events). There is an assumption that the resulting sum (or composition) of all the interactions will be equivalent to their sequence. But this ignores single-element interactions. Zippability only takes standardized behavioral reactions into account. It ignores the possibility of interacting elements in autonomous systems being able to process, learn, and adapt (e.g., bird flocks, insect swarms, and social systems). In mechanical systems, we are dealing with breakages rather than wear and tear.

In social systems, this kind of reductionism can lead one to disregard unwanted effects and side effects, which are often misunderstood as the price to pay for rendering the relevant phenomenon treatable.

*3.6. Social Systemic Reductionism*

Dealing with social systems, reductionism may be defined as the simple transposition of models and concepts used for physical and biological systems, reducing cognitive interaction to physical ones, essentially reducing cognitivism to forms, albeit of a certain complexity, of behaviorism.

Furthermore, as regards social systems, it is necessary to mention specific themes of reductionism [36].

In education, for instance, a critique of reductionism is very important to protect against misleading simplifications. The concept of reductionism has rarely been considered in education, where its use has been applied to very specific issues or as a vague term [37–39].

A similar situation is found when considering aspects of other kinds of social systems, such as in the field of management and healthcare, when reductionism consists of considering emergent and non-linearly caused properties to be organizational and functional.

Furthermore, as regards social systems, it is necessary to mention the introduction of specific systemic issues not coinciding with those mentioned above. A general, original theoretical systemic [40] approach was introduced by Niklas Luhmann's social theory and approaches to social systems [41].

His original systems theory focuses mainly on three topics: systems theory intended as societal theory, as communication, and as evolution theory. A system is identified by the boundary between itself and its environment. The boundary separates it from an indefinitely complex exterior. The "interior" of the system is thus a zone of so-called "reduced complexity", where interior communication occurs by selecting only a limited amount of all the information available outside.

The distinctive identity of each system is continuously reproduced in its communication and depends on what is considered meaningful or not. When a system fails to maintain this identity, it disappears as a system and dissolves back into the environment it emerged from. Luhmann terms this process of reproduction from elements previously selected from an over-complex environment as "autopoiesis", using the same term introduced by Maturana and Varela to identify the ability of systems to self-reproduce. Luhman considers social systems to be "operationally" closed since they use environmental resources, but they do not become part of the systems' operation.

Regarding reductionism, Luhmann wrote:

"For a long time, in sociology the representatives of an individualistic reductionism claimed to have achieved special access to the elementary, empirically graspable foundations of social life" [41] (p. 256).

and

"Every version of individualistic reductionism has encountered the objection that, as reductionism, it cannot be fair to the 'emergent' properties of social systems. We would object further that the issue is not even reductionism, but relating (in an extremely abbreviated way) to psychic rather than social systems." [41] (p. 257).

This reductionism relates mostly to emergentism in sociology. While in systemic emergence, the core problems considered above are, for instance, causation and reductionism, for Luhmann's theory, they are problems of meaning and self-reference [42].

At this point, we mention an interesting correspondence between the concepts of theoretical incompleteness and multiplicity introduced above, and what was considered by Luhman:

> "He wanted to avoid above all else the idea that one could capture 'the truth' or essence of modern society in one theoretical account. No theory, not even closed systems theory or autopoiesis, can have the last word or give an exclusive or true account of what society, in its totality, is and how it operates. One could even suggest that the first principle of Luhmann's sociology is that the possibility not only of seeing things differently but of society actually being different is always present. …What he wished to offer, therefore, was a social theory of social theories—a social theory which considered multiple ways of perceiving and understanding society." [43] (p. 1).

Regarding reductionism:

> "He fully realized that one could never completely escape reductionism, since any attempt to address and understand events socially necessarily involves selection, rejection and interpretation." [43] (p. 1).

*3.7. Reverse Reductionism*

Reverse reductionism is the opposite of unwarranted reductionism. It occurs when a phenomenon is forcibly modeled as a system. The phenomenon is adjusted and measurements are adapted so that a system appears to be present. Reverse reductionism can occur when available systemic models appear to be good approximations of unmodeled processes exhibiting the same unexplained regularities. Reverse reductionism loads a systemic nature onto some phenomenon under consideration.

For instance, an observer might recognize supposedly predominant systemic behaviors in (1) populations that exhibit Brownian motion, the irregular but continuous random thermal agitation of molecules in heated liquids or gasses caused by thermal energy, and (2) the turbulence found in smoke diffusion, terrestrial atmospheric circulation, and the mixed oceanic and atmospheric layers of ocean currents. The same occurs when recognizing systems in populations of arbitrarily inhomogeneous elements whose interaction is also only supposed and not detected.

Certain cases should, nonetheless, not be confused with reverse reductionism. These include the use of systemic models developed in a specific disciplinary context. This occurs in interdisciplinarity when approaches, problems, and solutions from one discipline are considered in terms of another by changing the meaning of variables or transforming one problem into another. The goal is to render such problems more easily treatable. This can take the shape of transforming geometric problems into algebraic ones, military problems into economic ones, energy consumption problems into social ones, medical problems into chemically treatable unbalances, and vice versa. In these cases, we can recognize the same systemic structure in different disciplinary contexts. This is a matter of using the same approaches and models. Such approaches and models are, however, applied in different contexts by changing the meanings of variables. Examples include Lotka–Volterra equations of prey–predator systems or chaotic climatic behavior being transposed onto economics. The adequacy of the systemic model's transposition decides its validity or its forcing. When the transposition is forced, we have reverse reductionism rather than interdisciplinarity (or forced interdisciplinarity).

## 4. Opaque Dark Systems

In this section, we consider phenomena whose systemic character is implicit and only detectable using appropriate variables, appropriate variations in scale, and an appropriate level of representation.

We now consider cases where the systemic nature of phenomenological dynamics and properties is hidden. This hiddenness might be due to the level of description assumed (i.e., scalarity) and the models an observer takes into account.

We use the term "opaque systems" metaphorically. Following Giuseppe Vitiello, "...opacity and transparency can be considered as a response of the body to the interaction with light, deriving, on the one hand, from the behavior of the body elementary components and structural properties, on the other from the intensity and frequency of light. One can have different responses to different light intensities and frequencies, also depending on whether the body has, for example, a crystalline or amorphous structure. Opacity and transparency are, therefore not intrinsic properties of bodies. They describe the way the body 'manifests' when using light as an instrument of observation" [44] (p. 42).

We use the term "dark systems" in a similarly metaphorical way. The term refers to the phenomenon of dark matter (dark because its presence is determined by its gravitational properties rather than its luminosity).

"Hidden", "opaque", and "dark systems" can be designated in two different ways: (a) constructively (i.e., supposed to explain and model phenomena) and (b) as discovered via sophisticated and dynamic scalarity changes.

Examples of opaque dark systems include (1) supposed systemic rules of phenomena at large temporal and spatial scales (as in astrophysics and anthropology), (2) the effects of the shadow economy, (3) pandemic effects (e.g., COVID-19), and (4) hidden natural systems at scale (e.g., Bènard rolls in fluid dynamics when considered in atmospheric phenomena) [45].

Are there approaches that might be suitable for discovering hidden opaque dark systems? Can we (idealistically) suppose the possibility of these kinds of system detectors?

Finally, and in reference to opaque dark systems, we mention how the Heisenberg Uncertainty Principle expresses the fact that it is not possible to determine simultaneously both the frequency and the time location of the waves' components with arbitrary precision. The Heisenberg Uncertainty Principle { XE "Uncertainty Principle" } has been generalized and related to the theory of "statistical estimates", introducing so-called Fisher information [46,47].

Very briefly, we can say that Fisher information is a measurement of the amount of information that an observable X carries about an unknown parameter of distribution that models X.

Let the level of Fisher information at the source have the value $I$. Let the observed level of Fisher information in the data have the value $J$. The Extreme Physical Information (EPI) principle states that $J - I$ = extremum. When this extremum is the minimum, the observed level is considered to match up with its source. The EPI principle is based on the idea that the observation of a "source" phenomenon is never completely accurate and the related information is necessarily lost from source to observation. Furthermore, intrinsic random errors are intended to define the distribution function of the source phenomenon.

Some authors [48–50] have considered Fisher information a grounding principle from which it is possible to derive physical laws.

The study of highly complex systems, for instance, cognitive and socio-economic ones, needs suitable generalizations of the Uncertainty Principle { XE "Uncertainty Principle" }. This is the case for highly complex systems characterized by the fact that every model adopted is, in principle, partial, having inevitable aspects of uncertainty, and can represent only certain particular features of the system under study, neglecting inevitably other features of no less importance, representing the opacity of systems.

The notion of open dark systems may provide a shortcut to the description of complex systems, and might be simply defined as complex systems/phenomena that are not easily understood or observed due to their inherent complexity, their lack of transparency, or the limited accessibility of information. These systems are often characterized by numerous interdependent and non-linear processes, making them difficult to analyze and predict using traditional reductionist approaches. Fisher information, generalizing the Uncertainty

Principle, may be suitable to seek out the uncertainties in complex systems like using Roy Frieden's Extreme Information Theory [49].

## 5. Theoretical Incompleteness and Logical Openness

In this section, we outline the concepts of theoretical incompleteness and logical openness. We will focus on their interrelations as theoretical research issues related to high levels of equivalence.

### 5.1. A Note on Theoretical Incompleteness

Theoretical incompleteness lies in facts like the following: (1) a single model is not sufficient to represent complexity, (2) system variables (degrees of freedom) vary in number and are continuously acquired, (3) non-equivalent properties are continuously acquired, and (4) systems can assume many equivalent states (as determined by fluctuations). Theoretical incompleteness is incompletable in principle. Examples of theoretical incompleteness include (1) the Uncertainty Principle in quantum mechanics (where accuracy in measuring one variable comes at the expense of another), (2) the Complementarity Principle in theoretical physics (between wave and particle natures), and (3) Gödel's incompleteness theorems [51].

As noted in the literature on complex systems, incompleteness is intended to be a theoretically necessary condition for emergence in a dynamic of equivalences. However, incompleteness is not sufficient for the assumption of coherence in the emergence of complex systems (e.g., for bird flocks and insect swarms) [8,20] (pp. 158–159), [52,53].

The emergence of complex systems requires theoretical incompleteness. Completeness can be thought of as the "enemy" of emergence. This is because it produces ruled contexts without a place for equivalences and multiple roles [8] (pp. 161–170), [20].

Regarding modeling theoretical incompleteness, we should mention elementary approaches and cases.

An elementary case consists of a system of equations containing fewer equations than variables (the number of the former is less than the number of the latter). This leaves one or more variables undefined in terms of the other variables. It is then an incomplete system.

One can also consider a case where the subsequent computational steps do not use the same system of equations. Instead, there are different (complete or incomplete) versions of the system. Equations come into play in different subsequent combinations. This is another example of the conceptual modeling of structural dynamics mentioned in Section 3.2.

We can think of theoretical incompleteness as taking place when systems of equations assumed to completely represent phenomena turn out to apply non-completely. In other words, they apply in non-completed combinatorial sequences that model aspects of quasi-ness and the structural dynamics of the phenomena under study. Examples include the incomplete variable sequences of deterministic chaos equations, of the Van der Pol and Lotka–Volterra models. Examples of theoretical incomplete systems include ecosystems and collective systems.

### 5.2. A Note on Logical Openness

The concept of logical openness has been introduced as an extension of thermodynamical openness and in contrast to logical closedness [54].

Logically closed modeling relates to thermodynamically closed systems. The evolution of such systems can be represented as follows:

i.   Formal and complete descriptions of relationships between a system's state variables are available.

ii.  Complete and analytically describable representations of interactions between a system and its environment are available.

Logical closedness is given by the fact that knowledge of these two points allows us to deduce all possible states a system can assume.

Returning to the discussion of emergence and functioning from Section 3.2, we might consider the logical closedness of a Laplacian clocklike world (understood as functioning rather than emergent). There is conceptual correspondence with the notion of computability when an algorithm (a complete computational procedure) is available. More generally, the reference is to the availability of a procedure intended to completely represent a certain process.

In contrast, logically open modeling or logical openness occurs when there is a violation of the above two points. We can think of logical openness as being given by the occurrence of an infinite or non-depleting number of degrees of freedom for a system (including its environment) [8] (pp. 47–51), [26]. This renders a representation of the system theoretically incomplete. This is the case for complex systems whose degrees of freedom (the system variables) vary in number and are continuously acquired. The corresponding modeling uses $n$ levels of representation characterized by (1) non-equivalence, (2) approaches for moving between levels (thereby allowing the simultaneous use of more than one level), and (3) the need to find comprehensive indexes, including measures of coherence, long-range correlations, network properties (e.g., the occurrence of small worlds), and properties of attractors in chaotic phenomena.

The incompleteness of logical openness is marked by (1) the use of a variable number of non-equivalent models, (2) the abductive and constructivist indefiniteness of $n$ levels of representation due to observer-generated representations and models, and (3) the relevant indexes employed [10] (pp. 157–165), like global ordering and polarization.

### 5.3. Logical Openness of Theoretical Incompleteness

Logical openness is a property of modeling [8] (pp. 45–51), [28], while theoretical incompleteness is a phenomenological property.

We now explore how logical openness relates to theoretical incompleteness. On the one hand, logical openness is a property of modeling incomplete theoretical phenomena. On the other hand, incomplete theoretical phenomena require logical openness in modeling. This duality is expressed in Table 1.

**Table 1.** Logical openness of theoretical incompleteness.

| Logical Openness | Theoretical Incompleteness |
|---|---|
| Modeling uses $n$ levels of representations characterized by the following:<br><br>- Non-equivalence,<br>- Approaches for moving between levels (thereby allowing the simultaneous use of more than one level),<br>- The need to find comprehensive indexes, including measures of coherence, long-range correlations, network properties (e.g., the occurrence of small worlds), and properties of attractors in chaotic phenomena. | Characterized by:<br>Dynamics of multiple equivalences,<br>A single model is not sufficient to represent the complexity of a system,<br>The system variables (degrees of freedom) vary in number and are continuously acquired,<br>Non-equivalent properties are continuously acquired,<br>Systems can assume many equivalent states (as determined by fluctuations). |

The validity domain of logical openness can be broader than that of theoretical incompleteness. This can occur when theoretical incompleteness is a quasi-property of systems that have mixed temporary behaviors alternating between completeness and incompleteness (e.g., flying bird flocks, bird flocks resting on the ground during the night, and bird flocks that gradually form from elements on the ground. A corporation acts a system only during the working hours, while some departments may act as assembly lines, i.e., as structured sets. An electronic system may constitute subsystems activated on request that are otherwise inactive).

Furthermore, a system can decide its level of openness (e.g., context sensitivity, interfacing, learning and usage of memory, and processing uses). This is facilitated by selecting and assuming limitations. This occurs in the processes of abduction, a hypothesis-inventing process that can be considered a kind of selection between suitable alternatives [55–58].

A generic instance of logical openness occurs when one deals with incompletely modellable phenomena. Such phenomena are (a) non-zippable into single equations (as in complex systems where the degrees of freedom vary in number) and (b) continuously replaced and acquired (as in collective systems). One requires a DYSAM-like approach based on the dynamical assumption and replacement of models. Specific cases relate to the non-proceduralization of certain phenomena [9]. The prototypical example of a procedure is an algorithm that allows for complete decidability. The general idea is that one needs step-by-step methods prescribing how and when to do something.

A sequence, the processing of the system states representing its behavior, may be considered algorithmic, i.e., Turing-computable. For instance, the system assumes a finite and limited number of configurations, degrees of freedom, and numbers of states, as found for logically closed systems. That is, the system's behavior is representable as a procedure.

Non-algorithmic, non-proceduralized sequences of the states of a system may assume unlimited configurations, degrees of freedom, and numbers of states, as determined for logically open systems of which the behavior is not representable as a procedure. Namely, it is not globally Turing-computable even if, possibly, it is locally Turing-computable, as for the sub-symbolic [26] processing of artificial neural networks (ANNs) and their deep machine learning using recurrent neural networks [59].

Furthermore, sequences of the states of a system cannot be proceduralized when, for instance, each step depends on:

-   fluctuations, the breaking of equivalences and symmetries, weak forces, external influences, and randomness in physical systems;
-   adaptation and learning in autonomous systems: in short, sequences of processes of emergence and self-organization are not summable and cannot be procedularized or zipped into the resulting one.

In such cases, typical of emergent complex systems, the process is not identically repeatable. The representation of phenomenological interactional mechanisms and process sequences are not analytical but adaptive and ongoing, as represented by natural computation [60,61].

Generally speaking, a procedure is intended to consist of generic instruction manuals. Examples include maintenance and job security procedures that are assumed to be complete (i.e., necessary and sufficient). Investment in security is a necessary condition for job security, but it cannot exhaust the problem of reaching 100% security. A sufficient condition (converging to completeness) constitutes managing job security as an emergent property, one that is continuously acquired by a working community (as in the Tenaris experience) [62]. Job security cannot be proceduralized. Examples of this in social organizations (e.g., corporations) include mixing well-structured aspects (e.g., wearing overalls and helmets, respect shifts and times) with undefined aspects (e.g., the psychological state of workers, the ability to communicate and cooperate). This leaves room for adaptation and learning. The system can also decide the level of openness (or completeness) by selecting and assuming limitations (as in abductive processes).

## 6. The Establishment of Predominant Systemic Domains

This concept was anticipated in related research about, for instance, the phenomena of systemic propagation [8] (pp. 170–175) and pre-properties (establishing compatibility) [8] (pp. 146–151). The concept of a field might be metaphorically helpful for identifying the concept of a domain. However, the contrasts and differentiations between them are suitable for specifying the general concept of a systemic domain.

In physics, each point of a field is intended to have the precise value of a definitory variable (e.g., electromagnetic or gravitational).

Systemic domains are spatial regions of possible options available to entering entities. An example of an available option is the admissible and compatible states of an entity that respect the relevant constraints and degrees of freedom (dimensions of the space).

A systemic domain is assumed to be given by configurations of variables, degrees of freedom, ranges of values, and admitted changes to an entering entity. This contrasts with fields (where only one value of the defining variable is available at each point). Systemic domains allow for multiple achievable, admissible, compatible, incomplete (partial or fuzzy), subsequent, equivalent, and non-equivalent choices [20] (pp. 6–10). The systemic domain is then taken to be represented over time by multiple authorized value intervals (which are available for the subsequent change). The system domain is intrinsically time-dependent. In other words, domains are dynamic and dependent on previous domains. Systemic domains are generated by the dynamics of a system, possibly even multiple, being active in the space (e.g., collective systems and ecosystems). For an incoming entity to be sensitive to the domain, it must be compatible with entities in the generating system. Consider the circular swarming of mosquitos around a light. The swarming imposes itself on any incoming mosquito but does not influence different insects and peripheral entities (e.g., plant particles blown by the wind). This also applies to a whirlpool, which makes incoming liquid flow the same way as the liquid already contained in the whirl. Such scenarios can be represented by the basin of an attractor [63].

System domains predominate in the face of non-destructive inputs. This occurs by way of factors like suitable compatibility, quantities, and timing. The properties of domain sequences can be considered to represent the evolution of the generative system. There is also the possibility of inserting an input with its own systemic domain to replace a current one. The former then becomes predominant (as a collective behavior in Brownian motion). Examples of possible applications include market behavior and crowd behavior. This approach can also increase the tolerance or robustness of a system by inducing the recovery of temporarily lost properties (e.g., coherence).

## 7. Conclusions

In this article, we introduced theoretical issues, some related to reductionism and others that could more properly be understood as systems research topics.

The issues related to reductionism were: (1) the reduction of sufficient conditions to necessary conditions, (2) the reduction of emergence to functioning, (3) the general linearizability of non-linear systems, (4) attempts to deduce the macroscopic from the microscopic, (5) the zippability of sequenced interactions into a single resulting interaction, (6) dynamic and theoretical mixed uses of reductionism and non-reductionism, and (7) reverse reductionism.

The issues related to theoretical research topics were: (8) opaque dark systems, (9) theoretical incompleteness as logical openness, and (10) systemic domains.

With reference to the theoretical nature of the topics covered, we take this opportunity to urge systems researchers not to exhaust their efforts in creating models and simulations. Admittedly, such devices can facilitate unique and important ways of studying complex systems (e.g., the study of stochastic chaos).

However, the increasing availability and amount of data, for instance, via simulations, is useless since data are meaningless if there are no related hypotheses and theories.

We mention how the generalization of the possibility of theory-less knowledge, with theory being replaced by suitable concordance, correlation, and correspondence, for instance, within Big Data, is untenable. Actually, it is possible to detect many cases of specific knowledge being produced without the availability of, or search for, theories, by using concordance, correlation, and correspondence in a data deluge, so-called Big Data [64], using data-driven approaches within very large databases [65]. However, in this regard, the assumption that correlation supersedes causation and theorization has been determined an improper generalization [33].

On the other side, theories should not be understood as the only ideal representations, effective in completely representing and explaining processes and phenomena. In this latter case, real data are considered coincident with solutions of suitably explicit symbolic formal models, i.e., suitable equations even intended as laws, such as in mechanics

and thermodynamics. We may consider examples where we deal with a lack of explicit, symbolic understanding. This is the case, for example, for complex phenomena and processes with multiplicities irreducible to each other, as considered above with DYSAM (see Section 3). This is a matter of non-zippability in formal representations such as processes of emergence, representations, and simulations carried out using sub-symbolic devices, for instance, neural networks whose properties cannot be analytically, symbolically explained but are attributable to the connection weights and levels in the context of equivalence (see Section 5.3).

This, however, does not affect the possibility of establishing effective approaches showing, for instance, the effectiveness of connectionism, evidence-based medicine, and homologous variables, emergence lacking comprehensive theory, the use and modification of networks, and the effects of uncertainty principles. More realistically we should consider future contexts mixing theoretically symbolic, non-symbolic, and data-driven approaches allowing soft theorization and incomplete theorization [65–67].

However, there should also be a focus on pending theoretical topics as the ones considered in this Special Issue *Theoretical Issues on Systems Science*. Research can then have a theoretical nature, which contrasts with the notion of implementing finalized research on demand [68,69]. This seems particularly relevant to contemporary research projects that are pursuant of specific (funded) objectives. We conclude by considering the centrality of theoretical research in systems science, currently disciplinarily oriented rather than inter-disciplinarily oriented, as theoretical research inevitably is.

**Funding:** This research received no external funding.

**Institutional Review Board Statement:** Not applicable.

**Informed Consent Statement:** Not applicable.

**Data Availability Statement:** No new data were created or analyzed in this study. Data sharing is not applicable to this article.

**Conflicts of Interest:** The author declares no conflict of interest.

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
