# Peer review of "Theoretical Reflections on Reductionism and Systemic Research Issues: Dark Systems and Systemic Domains"

_systems, doi:10.3390/systems12010002_

Round 1

Reviewer 1 Report

Comments and Suggestions for Authors

Very interesting theoretical paper.  Suggestions for improvement and completeness:

Extend the hard/mathematical/complex systems perspective to social systems with reference to Luhmann and other related systems theorists (and practitioners) from the anglosaxon schools of systems thought. This will strengthen the paper's interface with practice, and, of course, its practical relevance (I sense that there is an intention to link it with practice). It will also balance the "too Italian" flavour of the paper.  

Author Response

Dear Reviewer,

Thank you for your comments and suggestions.

Extend the hard/mathematical/complex systems perspective to social systems with reference to Luhmann and other related systems theorists (and practitioners) from the anglosaxon schools of systems thought. This will strengthen the paper's interface with practice, and, of course, its practical relevance (I sense that there is an intention to link it with practice). It will also balance the "too Italian" flavour of the paper. 

I inserted new text (new Section 3.6, pp. 10-11) and references highlighted in yellow.

Hope this new version is now acceptable for publication.

Thank you again.

Sincerely,

The author

Reviewer 2 Report

Comments and Suggestions for Authors

This is an unusual but interesting paper. The abstract sets the scene appropriately. Not all the keywords are used in the abstract. The introduction lacks a gentle entry into the topic. It begins with "The theoretical nature of this elaboration relates specifically to (a) abstract systemic representations and (b) properties suitable for applications, generalizations, and research approaches" but this is not elaborated on, so the non-knowledgeable reader will not be guided into the topic. And why is it particularly suitable for dealing with the complexity of phenomena and why does a reductionist approach ignore this? Also, what are opaque dark systems and reverse reductionism, and how does the latter work, and why should we be interested in it?

The discussion on reductionism is quite good, pointing to useful citations. I guess that polipathologies refers to multiple pathologies that interact, but it would have been nice for this to have been explained, and under what conditions pathologies arise. It also looks as though this might well be turning into a theory of quasiness, an interesting direction if explained adequately with enough elaboration.

The notion of dark systems provides a very nice shortcut to the description of complex systems, and might be simply defined as: complex systems/phenomena that are not easily understood or observed due to their inherent complexity, lack of transparency, or limited accessibility of information. These systems are often characterized by numerous interdependent and nonlinear processes, making them difficult to analyze and predict using traditional reductionist approaches. Interestingly, however, there are approaches in information theory that are capable of seeking out the uncertainties in complex reality, like Roy Frieden's Extreme Information Theory.

Overall, this is an interesting paper that I would  like to see develop further.

Comments on the Quality of English Language

Generally a good level of English, but with one of two curiosities that are easily solved.

Author Response

Dear Reviewer,

Thank you for your comments and suggestions.

I inserted new text and references highlighted in green.

This is an unusual but interesting paper. The abstract sets the scene appropriately. Not all the keywords are used in the abstract.

This is due to space limitations and the need for conciseness.

The introduction lacks a gentle entry into the topic. It begins with "The theoretical nature of this elaboration relates specifically to (a) abstract systemic representations and (b) properties suitable for applications, generalizations, and research approaches" but this is not elaborated on, so the non-knowledgeable reader will not be guided into the topic. And why is it particularly suitable for dealing with the complexity of phenomena and why does a reductionist approach ignore this? Also, what are opaque dark systems and reverse reductionism, and how does the latter work, and why should we be interested in it?

I added text on page 2

The discussion on reductionism is quite good, pointing to useful citations. I guess that polipathologies refers to multiple pathologies that interact, but it would have been nice for this to have been explained, and under what conditions pathologies arise. It also looks as though this might well be turning into a theory of quasiness, an interesting direction if explained adequately with enough elaboration.

I added text and a reference on page 4

The notion of dark systems provides a very nice shortcut to the description of complex systems, and might be simply defined as: complex systems/phenomena that are not easily understood or observed due to their inherent complexity, lack of transparency, or limited accessibility of information. These systems are often characterized by numerous interdependent and nonlinear processes, making them difficult to analyze and predict using traditional reductionist approaches. Interestingly, however, there are approaches in information theory that are capable of seeking out the uncertainties in complex reality, like Roy Frieden's Extreme Information Theory.

I added text and references on page 11 ad 12

Hope this new version is now acceptable for publication.

Thank you again.

Sincerely,

The author

Reviewer 3 Report

Comments and Suggestions for Authors

Dear author,

Thank you for this interesting work. While the topics investigated in the paper are worthy of discussion, I have some major comments about the work. Please see them below.

~p.1. Title~

I suggest making this title more descriptive toward the aim and conclusion of the study vis-a-vis systemic reductionism.

~p.1, section 1~

The first and concluding paragraphs of this section allude to an assumption that studies in systems thinking research do not focus enough on theoretical issues. This seems to be the research problem or research gap this study is addressing. This kind of a sweeping claim requires citation and further elaboration. 

Sections 5 and 6 are not linked to the topic of reductionism, which is the main topic of this paper. To make the paper more cohesive, it might be worthwhile to elaborate on the exploration of systemic reductionism rather than on the other topics, which may warrant their own separate paper/s.

It would help if you outlined your research question for each section, i.e., the question or issue you are trying to argue for or against. The subsequent section lack clarity in this regard.

~pp. 7-8, section 3.5~

The characters in numbers (1)-(5) were not legible in my PDF copy of the manuscript.

~p. 14, section 7~

The first paragraph suggests the intent of the study is to re-frame certain theoretical issues as systems research topics. This aim could be made clear in the "Introduction" section of this paper.

In the second paragraph, you seem to be implying that there isn't enough work being done on systems theory on the part of systems researchers. This kind of claim requires backing up with citation/s and it should also appear in your introduction (section 1).

~p.14, References~

It is important to make sure the author adds a contribution beyond their prior work in this paper. Given how many self-citations are included in the paper, this should be considered and described in the paper - how is the author building on their prior work to further explore topics in systemic reductionism?

Comments on the Quality of English Language

~p.1, paragraph 1~

"It is particularly" - what is "it" (the subject of the sentence) in this case? Unclear.

~p. 2, section 2, paragraph 2~

This sentence is structured in a way that makes it read as if the scientific method is what doesn't attempt to describe the nature of things, but I think you meant to say this about reductionism.

Please check the script for other sentences which may not be clear.

Author Response

Dear Reviewer,

Thank you for your comments and suggestions.

I inserted new text and references highlighted in blue.

 ~p.1. Title~

I suggest making this title more descriptive toward the aim and conclusion of the study vis-a-vis systemic reductionism.

I inserted a new version of the title

~p.1, section 1~

The first and concluding paragraphs of this section allude to an assumption that studies in systems thinking research do not focus enough on theoretical issues. This seems to be the research problem or research gap this study is addressing. This kind of a sweeping claim requires citation and further elaboration. 

I inserted new explicative text on pages 2 and 3.

The subject is then elaborated in the new text inserted on page 17

Sections 5 and 6 are not linked to the topic of reductionism, which is the main topic of this paper. To make the paper more cohesive, it might be worthwhile to elaborate on the exploration of systemic reductionism rather than on the other topics, which may warrant their own separate paper/s.

It would help if you outlined your research question for each section, i.e., the question or issue you are trying to argue for or against. The subsequent section lack clarity in this regard.

I inserted an explicative statement on page 2

I reworded the title of Section 6

~pp. 7-8, section 3.5~

The characters in numbers (1)-(5) were not legible in my PDF copy of the manuscript.

I reformatted. I hope they are legible now.

~p. 14, section 7~

The first paragraph suggests the intent of the study is to re-frame certain theoretical issues as systems research topics. This aim could be made clear in the "Introduction" section of this paper.

Please consider the new text on page 2 both in green and in blue

In the second paragraph, you seem to be implying that there isn't enough work being done on systems theory on the part of systems researchers. This kind of claim requires backing up with citation/s and it should also appear in your introduction (section 1).

I inserted an explicative text

~p.14, References~

It is important to make sure the author adds a contribution beyond their prior work in this paper. Given how many self-citations are included in the paper, this should be considered and described in the paper - how is the author building on their prior work to further explore topics in systemic reductionism?

The self-citations are introduced to provide documentation and references to the reader.

Comments on the Quality of English Language

~p.1, paragraph 1~

"It is particularly" - what is "it" (the subject of the sentence) in this case? Unclear.

I revised the text

~p. 2, section 2, paragraph 2~

This sentence is structured in a way that makes it read as if the scientific method is what doesn't attempt to describe the nature of things, but I think you meant to say this about reductionism.

I revised the text

Hope this new version is now acceptable for publication.

Thank you again.

Sincerely,

The author

Reviewer 4 Report

Comments and Suggestions for Authors

Dear editor,

I have read with attention the manuscript entitled “Theoretical Reflections on Reductionism and Related Systemic Research Issues” by G. Minati and submitted for publication in Systems.

The article proposes an overview of theoretical and conceptual issues related to the use of reductionist approaches to the characterisation of systems. More specifically, it takes a meticulous look at a set of conditions which ought to apply for reductionist programmes, broadly defined, to apply, and which may be either unknown or ignored by some practicioners in the community.

The article’s content does not appear to me as original or new but has the merit of putting together a list of valid theoretical objections against the blind use of reductionist approaches and narratives; the consequences of which often having the detrimental effect of oversimplifying a problem in general, and even more so when confronted to highly complex and interconnected structures such as ecological systems or the economy for example.

By laying out these theoretical areas in wants of further works beyond simple reductionism, the manuscript invites the community of experts in the theory of systems to explore novel research avenues to overcome these challenges. Consequently, this article’s scope appears to me as of interest to the readership of the Systems.

For these reasons, I am happy to recommend the publication of the article in Systems.

Remark:

There is a strange graphical glitch in the display of Eqs.(1)-(5).

Author Response

Dear Reviewer,

Thank you for your positive comments and for recommend the publication of the article in Systems.

I reformatted the Eqs. (1)-(5). I hope they are legible now.

Thank you again.

Sincerely,

Gianfranco Minati

Round 2

Reviewer 3 Report

Comments and Suggestions for Authors

Dear authors,

Thank you very much for your new manuscript and for making your edits to it. Please see my new comments below.

~Title~

Isn't reverse reductionism part of "Reductionism"? The subtitle is confusing for me to read if I pretend I have not read this manuscript previously.

~p.2, section1 ~

- I would suggest against using the term "blind use" which may come across as accusatory and instead opt for a term such as "misdirected use". After all, nobody would disagree that blind use of reductionism is a bad thing, but your thesis goes deeper than just saying this.

- Section 4 also concerns reductionism (reverse reductionism). It seems as if this section could also be included within the main scope of the paper. 

~pp. 2-3, section2

The explanation of the distinctions between reductionism and the scientific method is somewhat unclear. For example, the sentence "Reductionism must be distinguished from the scientific method which considers quantitative aspects, that is, collections of measurable quantities" reads as if reductionism does not deal with measurable quantities. Then later, it the sentence "Reductionism consists in considering that a finite and limited number of measurable quantities represents the essence of the apple" suggests the opposite. I think I realize what you meant to say once I think about it more, but it makes the reader's job more difficult.

~p.10-11, section3.6 ~

I think the passages and arguments in this section could be made clearer with improved phrasing. For example, the paragraph "Dealing with social systems, reductionism may be given by the simple transposition of models and concepts used for physical and biological systems, reducing cognitive interaction to physical one. Essentially by reducing cognitivism to forms, albeit of a certain complexity, of behaviorism. That is simplifying the level of representation." is confusing to understand in terms of subject and object. I think I realize what you meant to say once I think about it more, but it makes the reader's job more difficult.

~p. 12, section4.1

I think the passages and arguments in this section could be made clearer with improved phrasing. 

- There is a misplaced comma (red) and a missing comma (blue) in the following paragraph: "Finally, and in reference to opaque dark systems, we mention how the Heisenberg Uncertainty Principle, expresses the fact that it is not possible to determine simultaneously both the frequency and the time location of the waves’ components with arbitrary precision. The Heisenberg Uncertainty Principle has been generalized and related to the theory of ‘statistical estimates’, introducing the so-called Fisher Information [46, 47]. In very short the Fisher Information is a measurement of the amount of information."

- The phrase "In very short" is incomplete. In very short what?

- I am not clear on what the last part of this sentence means. The sentence structure with this number of commas could make it confusing to the reader: "Namely, highly complex systems are characterized by the fact that every model adopted is, in principle, partial and can represent only certain particular features of the system under study, neglecting inevitably other features of no less importance, representing the opacity of systems."

~p. 17, section7~

- At the top of this section, it would benefit the read to restate explicitly the objective/purpose of this inquiry and comment/reflect on the theoretical contribution of this work. 

I think the passages and arguments in this section could be made clearer with improved phrasing. 

- The following paragraph ends abruptly and then the next one is seemingly unrelated to it: "Given the above, we would like to urge systems researchers not to exhaust their efforts in creating models and simulations. Admittedly, such devices can facilitate unique and important ways of studying complex systems (e.g., the study of stochastic chaos).

As matter of fact, the increase in the amount of data, for instance through simulations, is useless since data are meaningless if there are no related hypotheses and theories."

I think you might have meant "however" rather than "a matter of fact". Even then, I am not clear on the connection between the two paragraphs.

- I found the argument concerning theory-less approaches somewhat unclear and incomplete. Since data-driven approaches do produce knowledge, the phrase "mathematically wrong" ought to be elaborated a bit more or the relevant section with more detail referred to. Also, the use of the phrase "matter of fact" seems oddly placed when making a theoretical argument.

Comments on the Quality of English Language

I suggest going through the entire manuscript for improved phrasing, not only over the newly added/edited passages. You should look for simplifying run-on sentences with many commas. For more detailed comments and examples, please see my other comments and suggestions.

Author Response

Dear Reviewer,

Thank you for your comments and suggestions.

I inserted new text and references highlighted in blue.

Dear authors,

Thank you very much for your new manuscript and for making your edits to it. Please see my new comments below.

~Title~

Isn't reverse reductionism part of "Reductionism"? The subtitle is confusing for me to read if I pretend I have not read this manuscript previously.

I removed “Reverse reductionism”

~p.2, section1 ~

- I would suggest against using the term "blind use" which may come across as accusatory and instead opt for a term such as "misdirected use". After all, nobody would disagree that blind use of reductionism is a bad thing, but your thesis goes deeper than just saying this.

done

- Section 4 also concerns reductionism (reverse reductionism). It seems as if this section could also be included within the main scope of the paper. 

Thank you, done

~pp. 2-3, section2

The explanation of the distinctions between reductionism and the scientific method is somewhat unclear. For example, the sentence "Reductionism must be distinguished from the scientific method which considers quantitative aspects, that is, collections of measurable quantities" reads as if reductionism does not deal with measurable quantities. Then later, it the sentence "Reductionism consists in considering that a finite and limited number of measurable quantities represents the essence of the apple" suggests the opposite. I think I realize what you meant to say once I think about it more, but it makes the reader's job more difficult.

Yes, right, I removed the controversial and unnecessary sentence.

~p.10-11, section3.6 ~

I think the passages and arguments in this section could be made clearer with improved phrasing. For example, the paragraph "Dealing with social systems, reductionism may be given by the simple transposition of models and concepts used for physical and biological systems, reducing cognitive interaction to physical one. Essentially by reducing cognitivism to forms, albeit of a certain complexity, of behaviorism. That is simplifying the level of representation." is confusing to understand in terms of subject and object. I think I realize what you meant to say once I think about it more, but it makes the reader's job more difficult.

I removed the last, possibly confusing sentence

~p. 12, section4.1

I think the passages and arguments in this section could be made clearer with improved phrasing. 

- There is a misplaced comma (red) and a missing comma (blue) in the following paragraph: "Finally, and in reference to opaque dark systems, we mention how the Heisenberg Uncertainty Principle, expresses the fact that it is not possible to determine simultaneously both the frequency and the time location of the waves’ components with arbitrary precision. The Heisenberg Uncertainty Principle has been generalized and related to the theory of ‘statistical estimates’, introducing the so-called Fisher Information [46, 47]. In very short the Fisher Information is a measurement of the amount of information."

I cannot see the colors of your comments. However, I inserted changes.

- The phrase "In very short" is incomplete. In very short what?

I completed

- I am not clear on what the last part of this sentence means. The sentence structure with this number of commas could make it confusing to the reader: "Namely, highly complex systems are characterized by the fact that every model adopted is, in principle, partial and can represent only certain particular features of the system under study, neglecting inevitably other features of no less importance, representing the opacity of systems."

I inserted explicative text

~p. 17, section7~

- At the top of this section, it would benefit the read to restate explicitly the objective/purpose of this inquiry and comment/reflect on the theoretical contribution of this work. 

I inserted new text

I think the passages and arguments in this section could be made clearer with improved phrasing. 

- The following paragraph ends abruptly and then the next one is seemingly unrelated to it: "Given the above, we would like to urge systems researchers not to exhaust their efforts in creating models and simulations. Admittedly, such devices can facilitate unique and important ways of studying complex systems (e.g., the study of stochastic chaos).

I inserted a connection sentence

As matter of fact, the increase in the amount of data, for instance through simulations, is useless since data are meaningless if there are no related hypotheses and theories."

I think you might have meant "however" rather than "a matter of fact". Even then, I am not clear on the connection between the two paragraphs.

I rephrased

- I found the argument concerning theory-less approaches somewhat unclear and incomplete. Since data-driven approaches do produce knowledge, the phrase "mathematically wrong" ought to be elaborated a bit more or the relevant section with more detail referred to. Also, the use of the phrase "matter of fact" seems oddly placed when making a theoretical argument.

I rephrased

Comments on the Quality of English Language

I suggest going through the entire manuscript for improved phrasing, not only over the newly added/edited passages. You should look for simplifying run-on sentences with many commas. For more detailed comments and examples, please see my other comments and suggestions.

I will proceed once the final text will be available

Hope this new version is now acceptable for publication.

Thank you again.

Sincerely,

The author

Round 3

Reviewer 3 Report

Comments and Suggestions for Authors

Dear author,

Thank you for making the edits to your manuscript.

I suggest the title "Theoretical Reflections on Reductionism and on Other Systemic Research Issues -Dark Systems and Systemic Domains" (or something similar). I have no other comments.

Comments on the Quality of English Language

I have made more or less the same comments on previous versions of this manuscript. The phrasing of some sentences should be improved for clarity. I suggest having a native-level English speaker go through the manuscript to improve the phrasing. The authors have already said they would go through this in the final version of the manuscript.